

# BUOYANCY-DRIVEN EFFECTS ON TURBULENT DIFFUSIVITY INDUCED BY A RIVER PLUME IN THE SOUTHERN BRAZILIAN SHELF

Rafael André Ávila[1] and Paulo H.R. Calil[1]

[1]Laboratório de Dinâmica e Modelagem Oceânica (DinaMO), Instituto de Oceanografia, Universidade Federal do Rio Grande

**Correspondence:** Rafael André Ávila (rafael.avila@furg.br)

**Abstract.** Freshwater plumes are important flow structures that influence the dynamics and water properties of coastal regions and continental shelves. Turbulence in plume regions is mainly driven by shear instabilities at the interface between plume and oceanic waters, which, in turn, depend on the geometry and outflow of a specific plume region. The Southern Brazilian Shelf presents a highly variable hydrographic distribution modulated by the seasonal wind variation and the freshwater discharge from the La Plata River estuary, which has a significant impact on the continental shelf circulation. This buoyant plume creates strong density gradients and interacts with local water masses resulting in a complex hydrographic pattern. In this study, high resolution hydrography and microstructure measurements were obtained in order to verify the effect of freshwater stratification on vertical mixing in this highly dynamic continental shelf. Results show that the plume is highly stable at southern portions of the shelf, as density displacements, or Thorpe displacements, $\delta_T$, heat diffusivity, $K_T$, buoyancy flux, $B_f$, and density gradient ratio, $R_\rho$ are reduced when compared to the northern areas. Moreover, hydrographic data suggests that the large-scale La Plata River plume has a dynamic mid-field region due to instabilities generated when reaching the shelf break.

## 1   Introduction

Freshwater plumes occur in a large range of sizes and shapes primarily determined by the river or estuary outflow (Simpson, 1997; Horner-Devine et al., 2015). Depending on their magnitude, freshwater plumes have a major influence on the dynamics, water properties and circulation patterns of coastal regions (e.g., Peters, 1997; Hetland, 2005; MacDonald et al., 2007). Although the freshwater discharge is the most important parameter (and possibly the most variable) in the structure of a plume, tidal amplitude, ocean currents, wind stress, coastline shape and local bathymetry can also affect their dynamics. Moreover, plumes with horizontal length scales larger than the local Rossby radius of deformation are also affected by Earth's rotation (e.g., Pimenta et al., 2005). The morphological and physical variability within plume regions will affect their turbulent characteristics (Horner-Devine et al., 2015). The prototypical type has three distinct regions in terms of mixing: the near, mid and far-field regions. In this type of plume, circulation is forced by a narrow channel and, sometimes, the Coriolis force. Intense mixing is prone to occurs in the near-field region (e.g., MacDonald et al., 2013), the portion of initial expansion where the momentum of the plume surface layer dominates over its buoyancy, creating a strong vertical shear that results in intense mixing. Between the near and far-field, the mid-field region is where the Earth's rotation affects the plume advection, restraining the



offshore spreading and turning it to the right (left) in the Northern hemisphere (Southern hemisphere). A stratified-shear flow may occur due to deceleration in the mid-field, increasing frontal mixing. Given the two-layer system formed by the encounter of plume and ocean waters, mixing in freshwater plumes occurs through diapycnal transport of buoyancy and momentum at the plume interface (Ivey et al., 2008). Therefore, quantification of mixing is given by the turbulent transport of momentum

and buoyancy fluxes (Horner-Devine et al., 2015). Diapycnal mixing is primarily driven by Kelvin-Helmholtz instabilities, although their development depends on the magnitude of stratification and advection, which in turn is driven by buoyancy and wind shear (e.g., Lentz, 2004; Wang et al., 2015). Usually, buoyant plumes have a small aspect ratio and thus vertical mixing is generally considered to be dominant over the horizontal turbulent fluxes.

The Southern Brazilian Shelf (hereafter SBS), is the continental shelf area located between the latitudes of 28°S and 34°S,

from Chuy to the vicinities of Sta. Marta Cape, in the southwestern limb of the South Atlantic Ocean (Fig. 1). Hydrographic characteristics in the region are modulated by the seasonal wind, which is NE during spring and summer and SW during autumn and winter (Soares and Möller, 2001). The primary source of freshwaters in the SBS is the La Plata River estuary, with an average discharge of, approximately, $22500 \, \mathrm{m^3 s^{-1}}$, which has a significant impact on the local dynamics as it creates strong horizontal and vertical density gradients (Guerrero et al., 1997). The estuary outflow as a coastal plume, namely the Plata Plume

Water (PPW), spreads along the coasts of Argentina, Uruguay and Brazil mixing with water masses from the Malvinas and Brazil currents (Möller et al., 2008). The plume interacts with the waters transported southward by the Brazil Current, namely Tropical Water (TW) and the South Atlantic Central Water (SACW). The mixing between PPW and TW forms the Subtropical Shelf Water (STSW), which occupies subsurface layers at the interface between the plume and oceanic waters. The northward flow of PPW is nearly geostrophic, following along the edge of the coastline as a trapped Kelvin Wave (Pimenta et al., 2005).

Although the spatial distribution of PPW over the continental shelf is driven by the La Plata River discharge and the Earth's rotation, the seasonal wind pattern affects the northward/eastward displacement of the plume (e.g., Soares and Möller, 2001; Möller et al., 2008). SW winds enhance the northward spreading, causing PPW to reach the vicinities of the Sta. Marta Cape. This northward flow carried by SW winds induce a relatively slow but highly energetic coastal current reported in the literature as the Brazilian Coastal Current (BCC) (De Souza and Robinson, 2004). Conversely, NE winds act to reduce the northward

displacement of the plume and spreads it towards the shelf break through Ekman transport, allowing a larger presence of warm salty waters near coastal areas and inducing upwelling near Sta. Marta Cape (Campos et al., 2013).

In this study we verified the buoyancy-driven effects on vertical mixing in a shallow, highly dynamic continental shelf. High resolution hydrography and microstructure measurements were obtained in order to estimate density displacements, or Thorpe displacements, $\delta_T$, heat diffusivity, $K_T$, buoyancy flux, $B_f$ and density gradient ratio, $R_\rho$, in areas influenced by the La Plata

River plume in two distinct locations in the SBS. The measurements herein are the first of this kind in the area and add relevant information regarding the dynamics and stability of the plume in this important region of the southwestern limb of the Atlantic Ocean.





## 2   Material and methods

The data was obtained in the SBS in two cruises of opportunity onboard R.V. *Atlântico Sul*, from Universidade Federal do Rio Grande, Brazil. The first one was carried out from June 2-11 and the second one from June 30 to July 8, both in 2015. The cruise tracks were planned as transects from the coast to the shelf break, according to time limitations and sea conditions.

Weather conditions were mostly calm during both cruises, except for a storm which occurred in the second cruise prior to the last station, on July 3. No significant changes in the La Plata River discharge and sea surface temperature were observed from the first to the second cruise.

The hydrographic data was collected with an OCEANSCIENCE Underway CTD (UCTD), equipped with a SeaBird probe, during the first cruise, obtained in a cross-shelf transect carried out near the location of Rio Grande (Fig. 1). Estimates of

velocity gradients were obtained using horizontal density gradients from the UCTD. In a geostrophic flow, velocity gradients can be obtained from the thermal wind balance. The horizontal density gradients were averaged into a grid with a horizontal resolution of 2 km and a vertical resolution of 1 m. This resolution was chosen in order to retain horizontal gradients akin to the geostrophic balance while removing smaller scale disturbances (Yankovsky, 2006). The resulting velocity gradients were used to estimated the gradient Richardson Number, $Ri_g = N^2/(\frac{\partial v}{\partial z})^2$, where $N$ is the buoyancy frequency,

$N = [(g/\rho_0)(\partial \rho/\partial z)]^{1/2}$.

The microstructure data was collected with a vertical microstructure profiler (VMP-250) during the second cruise, in two cross-shelf transects close to the locations of Albardão (AL) and Sta. Marta Cape (SM). Profiles were made in 13 stations, 8 in the AL and 5 in the SM transect (Fig. 1). During the survey, the instrument was equipped with a SBE7 micro-conductivity and a FP07 thermistor, which measured conductivity and temperature, respectively, at a 512 Hz sampling rate. The instrument

operated in downcast mode, with drop speeds varying between $\sim$0.2-1.4 ms$^{-1}$. The profiles were processed with 1 second FFT (Fast Fourier Transform) in 512 data point segments with 50% of overlap.

The microscale temperature allowed us to calculate the rate of loss of temperature variance ($\chi_T$), a measurable parameter that describes the effect of turbulence on the temperature field (Steinbuck et al., 2009). $\chi_T$ is obtained from the integration of the small scale temperature gradient spectrum ($\Phi_T$),

$$\chi_T = 6D_T \int_{k_1}^{k_2} \Phi_T(k) \, dk \quad [K^2 s^{-1}], \tag{1}$$

where $D_T$ is the molecular thermal diffusivity, equal to $1.4 \times 10^{-7} ms^{-1}$.

The magnitude of convection and temperature overturns due to heat fluxes can be expressed by the buoyancy flux ($B_f$),

$$B_f = \frac{g\alpha F}{\rho_0 c_p} \quad [m^2 s^{-3}], \tag{2}$$

where $g$ is gravity, $\alpha$ is the heat expansion coefficient, $\rho_0$ is the average density and $c_p$ the specific heat at constant pressure,

equal to $3.98 \times 10^3$ kgm$^2$/Ks$^{-2}$. As described in Thorpe (2007), $F$ is the turbulent heat flux, which is approximately equal to



$-\rho c_p K_T \langle (dT/dz) \rangle$, where $K_T$ is the eddy diffusivity of heat, calculated from the *Osborn-Cox* model (Osborn and Cox, 1972) as

$$K_T = \frac{\chi_T}{2\langle (dT/dz)^2 \rangle} \quad [m^2 s^{-1}]. \tag{3}$$

From the microscale density profiles we obtained the *Thorpe* displacements, $\delta_T$, a signature of mixing in stratified flows in the form of Kelvin-Helmholtz instabilities (Smyth and Moum, 2000). $\delta_T$ is the vertical distance that a water parcel must be moved adiabatically to restore stability (Mater et al., 2013). The displacements were calculated from the observed instantaneous microscale density profiles from the VMP-250. Discrete density measurements are monotonically sorted to give a gravitationally stable profile. The respective pressure associated with the sorted density profile is then subtracted from the local pressure, so that $\delta_T$ is given in metres.

To quantify and determine double-diffusive processes in the profiles we calculated the density gradient ratio, $R_\rho$,

$$R_\rho = \frac{(\alpha dT/dz)}{(\beta dS/dz)}, \tag{4}$$

with the microscale temperature and salinity gradients obtained with the VMP-250. $\alpha$ is the thermal expansion rate and $\beta$ is the haline contraction rate. $\alpha dT/dz$ is the contribution of the mean vertical temperature gradient to the vertical density gradient and $\beta dS/dz$ is the corresponding contribution of the mean vertical salinity gradient (Kelley et al., 2003). Double-diffusive convection (DDC) may be developed within the range $0 < R_\rho < 2$. A diffusive regime is favored when $0 < R_\rho < 1$, while a salt-finger regime is developed when $1 < R_\rho < 2$.

## 3 Results

### 3.1 Field observations

#### 3.1.1 Hydrographic observations: first cruise (early June of 2015)

High resolution transects of salinity (Fig. 2, panel (a)) and temperature (panel (b)) highlight the variability associated with the presence of the plume. PPW is observed at the surface along the entire transect, reaching the vicinities of the shelf break. Our observations show that the plume interface follows along the $\sim 23.5$ kgm$^{-3}$ isopycnal. The liftoff point of the plume, i.e., the location where the bottom attached salinity front loses contact with the bottom, is visible at the inner shelf region, approximately 50 m depth and 35 km from the starting position of the transect (about 75 km from the coast). TW and SACW are also observed in the final portion of the transect, as it reaches the shelf break. Warm and salty TW is associated with the southward flow of the Brazil Current (Soares and Möller, 2001) and SACW is ubiquitous on the Brazilian continental shelf from the northern portions of the South Brazilian Bight (e.g., Castro, 2014; Cerda and Castro, 2014) to the Brazil-Malvinas Confluence (e.g., Piola et al., 1999; Möller et al., 2008). Colder waters from the SACW may intrude the shelf region because



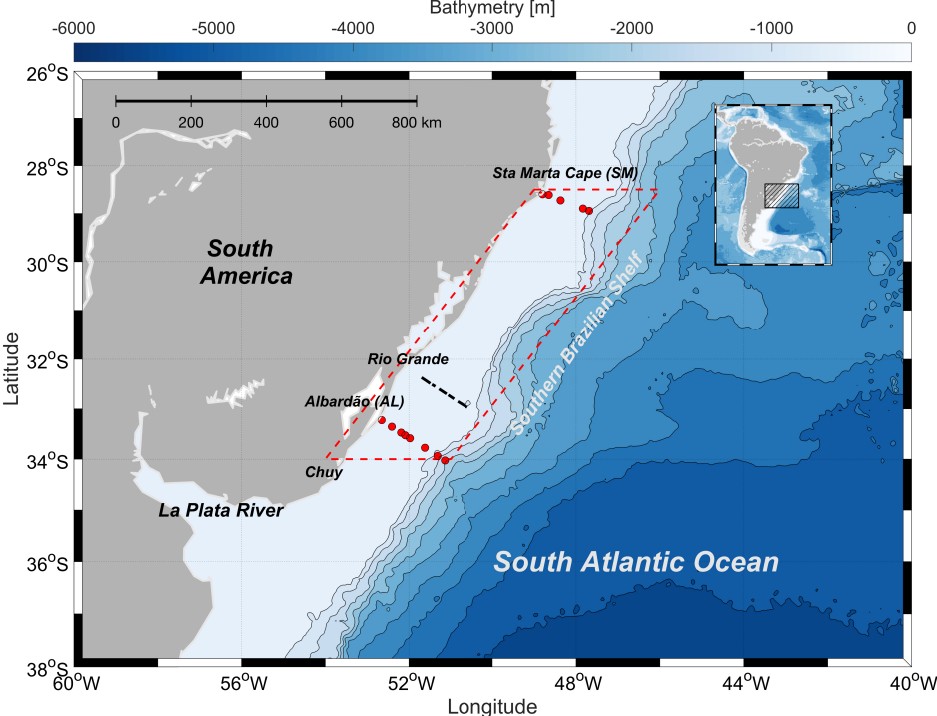

**Figure 1.** The Southern Brazilian Shelf region and vicinities. The red dashed box is the corresponding area of the shelf. The black thick lines are the UCTD casts made during the first cruise and the red dots are the VMP-250 profiles made during the second cruise.

of instabilities associated with the meandering of the Brazil Current as it flows along the shelf-break (Lima et al., 1996). Along the boundary between PPW and TW, STSW is observed, formed due to mixing between those water masses.

The transect of buoyancy frequency (Fig. 3, panel (a)) shows the strong stratification induced by the plume in this portion of the SBS. Values up to $0.05\ s^{-1}$, in the mid-section at the continental shelf, and close to the surface at the shelf break, are

5 observed. The vertical shear of the horizontal, geostrophic velocities (panel (b)) is mostly weak along the transect, but it is enhanced at the shelf break close to the surface, suggesting strong advection (northward at the southern hemisphere) at the most offshore portion of the transect. This strong shear led to the reduction of $Ri_g$ at this location (panel (c)). Low values are observed from the surface to the bottom of the profiles, suggesting that instabilities are occurring at the shelf break. This is consistent with what appears to be a vertical intrusion observed at, approximately, 50 m depth, which is clearer on the

10 temperature data (c.f., Fig. 2, panel (b)). Warm and saltier water on the shallow side rises while fresh and colder water on the deeper side of the shelf sinks, suggesting a vertical overturning at a sloping bottom, followed by a geostrophic jet (e.g., Wang, 1984). Over continental shelf, $Ri_g$ increases substantially in the mid transect section, where layers of enhanced $N$ are observed,





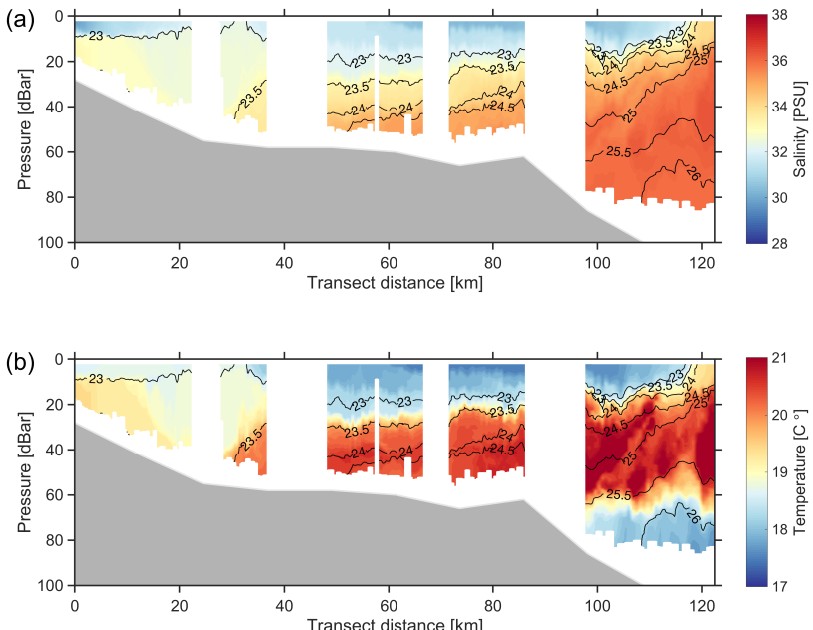

**Figure 2.** High resolution hydrographic transects of (a) salinity and (b) temperature obtained with the UCTD. The black lines are the density contours in $\mathrm{kgm}^{-3}$.

suggesting strong stability. Low values of gradient Richardson number are also found at the shallow inner shelf portion, where stratification is weakened due to the shallow depth.

### 3.1.2 Microstructure observations: second cruise (early July of 2015)

The frequency of occurrence of $K_T$ and $R_\rho$ for both transects were clustered into continental shelf and shelf break data due

to differences regarding the presence of PPW. For the AL transect, $K_T$ was nearly normally distributed (Fig. 4, panel (a)), being similar for continental shelf and shelf break. A storm occurred at the end of the AL transect, which enhanced mixing and surface $K_T$ in the shelf break profiles, increasing the occurrence of larger values. The averages of each distribution were similar, being slightly larger at shelf break. For the SM transect (Fig. 5 panel (a)), on the other hand, the distributions were unequal for both areas, being nearly bi-modal. This happened because of the smaller amount of plume waters in the continental

shelf off Sta. Marta Cape, allowing increased levels of vertical diffusivity in the shelf region, on average, nearly one order of magnitude greater than at the shelf break.

The $R_\rho$ distributions (panel (b) in Fig. 4 and Fig. 5) evidence the differences between continental shelf and shelf break. In the shelf break profiles, DDC can be developed by salt-fingers, where warmer and saltier TW lies over colder and fresher SACW. In the continental shelf, conversely, where colder and fresher waters from the plume stand above relatively warm and

15 salty oceanic waters, DDC can occurs due to a diffusive regime. Actually, the distribution mode of $R_\rho$ in the continental shelf



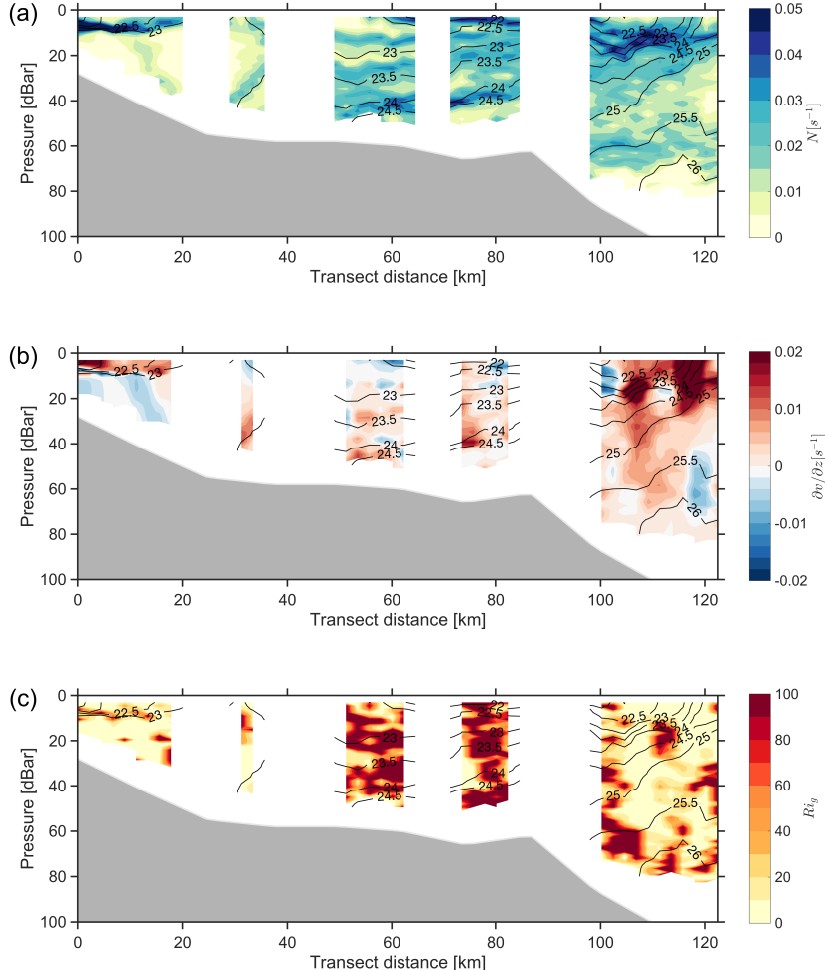

**Figure 3.** Transects of (a) $N$, (b) $\partial v / \partial z$ and (c) $Ri_g$. The black lines are the density contours in kgm$^{-3}$.

for the AL transect suggests that diffusion is unlikely to occur since the most frequent values were around 0.01. This is result of the strong stratification induced by plume waters in the southern portion of the SBS. For comparison, in the SM transect the mode is around 0.28, which is more likely to yield diffusivity.

The $N$ profiles in the AL transect (Fig. 6, panel (a)) match the hydrographic observations from the UCTD, with values up
5  to 0.05 $s^{-1}$ at the plume interface, the well stratified layer from stations AL03 to AL06, around 25 m depth. Below the $\sim$25 kgm$^{-3}$ isopycnal stratification is considerably reduced and the surface mixed layer (SML) sinks significantly beyond the shelf break. This significant sinking of the SML depth is also related to a storm that occurred near the location of station AL08. The strong vertical shear induced by this event caused the SML to reach nearly 135 m depth at this station. At the location of the





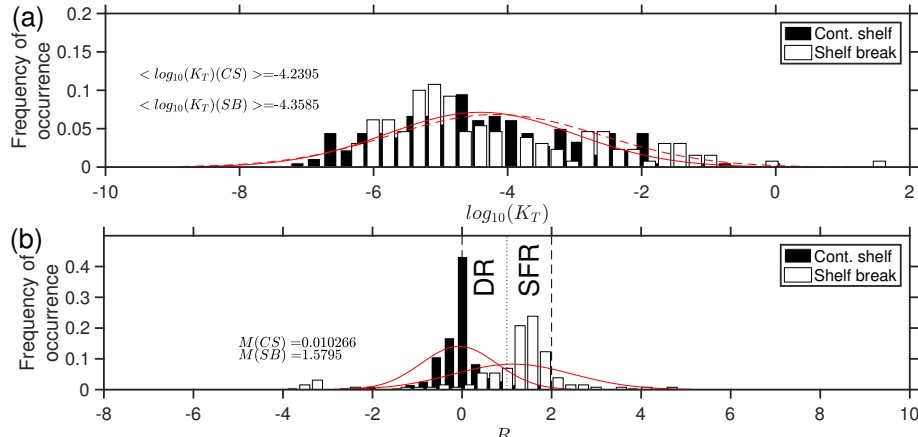

**Figure 4.** Frequency of occurrence of $K_T$ and $R_\rho$ for the **AL** profiles. In panel (b), DR stands for diffusive regime and SFR for salt-finger regime. The vertical dashed lines in (c) refer to the thresholds within each type of DDC regime is prone to occur. The red lines are the log-normal probability functions fitted to the data (full for continental shelf, dashed for shelf break). The ensemble average of each distribution is displayed in the figures.

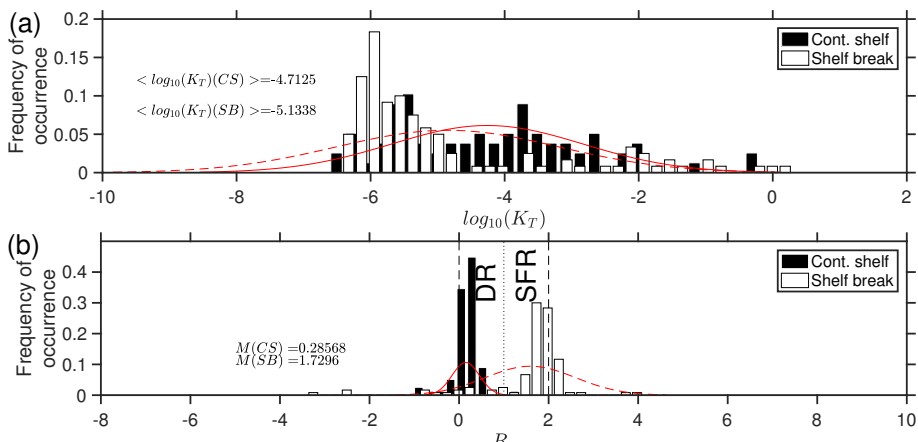

**Figure 5.** Same of Fig 4 for the **SM** profiles.

SM transect, a substantial reduction in $N$ occurs (Fig. 7, panel (a)) and a retreat of the plume interface towards the coast is observed.

Salinity and temperature gradients for both transects (Fig. 6 and 7, panels (b) and (c)) highlight the freshwaters over the continental shelf, as well as the stronger gradients in the southern transect, specially of salinity. Because of the buoyancy-driven salinity, temperature increases with depth (within the plume layer), an ubiquitous feature for all locations where PPW is observed (e.g., Castello and Möller, 1977).





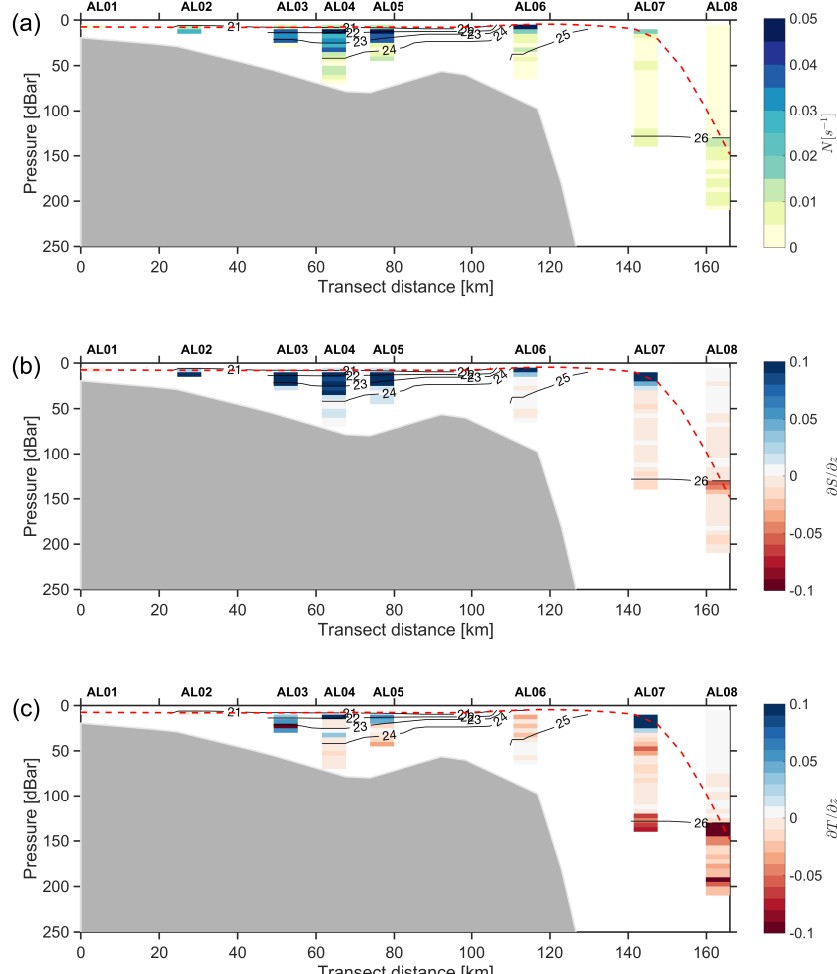

**Figure 6.** Vertical profiles of (a) $N$, (b) $\partial S/\partial z$ and (c) $\partial T/\partial z$ calculated from microstructure data for the **AL** transect. The black lines are the density contours in kgm$^{-3}$ and the red dashed line is the SML depth.

In the AL transect, $K_T$ and $B_f$ (Fig. 8, panels (a) and (b)) are significantly increased within the SML at AL08. In fact, heat diffusivity is very large, up to $\mathcal{O}(-2)$ m$^2$s$^{-1}$. Both heat diffusivity and buoyancy flux are reduced towards the continental shelf, mainly at the plume interface. At this location a reduction of nearly four orders of magnitude, when compared to the SML in AL08, is observed. In the SM transect, $K_T$ and $B_f$ are also large in the SML at the shelf break profiles (Fig. 9, panels (a) and (b)). But since no strong mixing event occurred, these relatively large values may be associated with surface convection, as the SML depth is nearly the same for both profiles, around 75 m. Heat diffusivity and buoyancy flux are reduced in the continental shelf, but not as much as observed in the AL transect. In fact, at the plume interface, no clear reduction in diffusivity is observed, as stratification is not as strong as in the location of the AL transect.





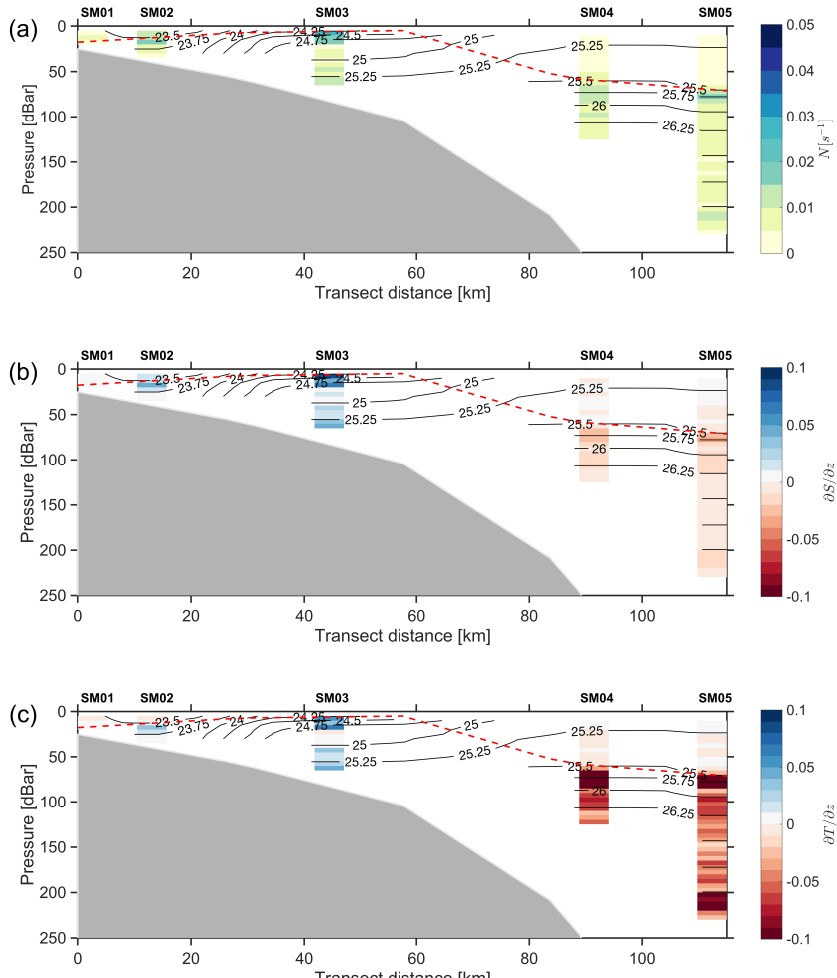

**Figure 7.** Same of Fig. 6 for the **SM** transect.

### 3.2 Effects of freshwaters stratification on mixing

In general, turbulent mixing in freshwater plumes is driven by shear-stratified flow instabilities in the form of KH billows within the stratified interface [Stacey et al. (1999)]. The stratification induced by PPW and the high $Ri_g$ values found on the UCTD data at the southern inner shelf implies low shear variance. Hence, the large-scale La Plata River plume reduces the

5    level of mixing expected for a shallow continental shelf. In large-scale ROFI (Regions Of Freshwater Influence) systems, the spreading outflow tends to stratify the water column, opposed by stirring due to tides, waves and winds (Simpson, 1997). This highly stable outflow inhibits the mixing at the interface due to the positive buoyancy input from low salinity waters, which exceeds the buoyancy loss from surface cooling. In the case of La Plata River plume, the induced buoyancy is very strong,





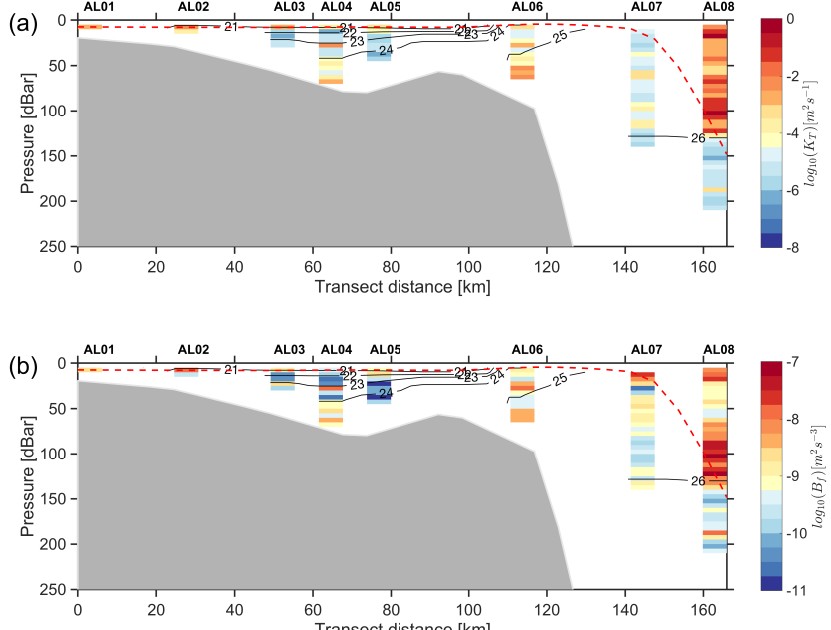

**Figure 8.** $Log_{10}$ profiles of (a) $K_T$ and (b) $B_f$ for the **AL** transect. The black lines are the density contours in $kgm^{-3}$ and the red dashed line is the SML depth.

so this ROFI system remains stratified even when the other portions of the shelf are mixed. Atmospheric forcing in the SBS, specially in the southern area, is restricted to the surface due to the effective isolation from bottom layers promoted by the freshwater-driven stratification (Zavialov et al., 2002). This explains the very shallow SML depth observed within the plume and the magnitude of $N$ observed in the plume interface.

Figure 10 shows $\delta_T$ calculated from the microstructure density profiles made on the SBS. Displacements are reduced in the AL profiles, more specifically, at the corresponding location of the plume interface. The density range of $\sim$21-24 $kgm^{-3}$, where buoyancy frequency reaches values up to $0.05\,s^{-1}$, is a strong and stable interface that inhibits mixing on the continental shelf and prevents the generation of shear instabilities. At the location of the SM transect, the density front is weakened due to higher dilution with oceanic waters. Consequently, it is more susceptible to shear-stratified turbulence, as seen by the increase in $\delta_T$. The presence of density overturns in strongly stratified flows is an indication of the amount of TKE within a flow, rather than its rate of dissipation (Mater et al., 2013). The reduced overturns at the plume interface in the AL region suggests that the flow, although energetic as an estuarine outflow, is stable.

Near-field river plumes are usually characterized by large turbulence and mixing, commonly found at estuaries with narrow channels (e.g., Luketina and Imberger, 1989; MacDonald et al., 2007, 2013). The wide the La Plata estuary, however, will rarely generate large velocity differences between upper and lower layers and shoaling as the plume leaves the estuary. As a consequence, a near-field region will not exist in the La Plata River plume and no large turbulence is expected to occur near





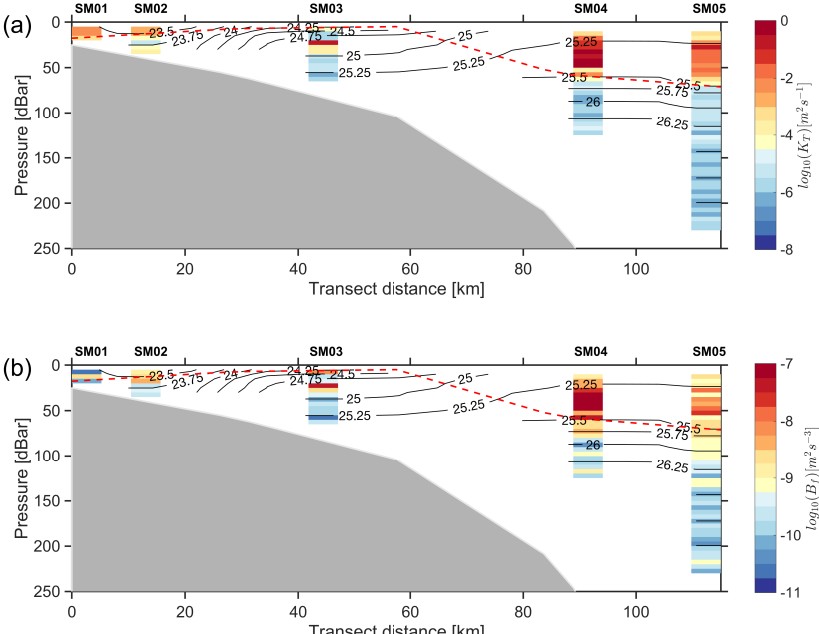

**Figure 9.** Same of Fig. 8 for the **SM** transect.

the source region. Due to its large spatial scale, which is larger than the internal Rossby radius (Pimenta et al., 2005), the La Plata River plume behaves as a far-field river plume, where motions are mainly driven by the Coriolis force and the wind shear away from the source region. When winds and local currents are not sufficient to advect the plume offshore, the far-field plume forms a geostrophic coastal current that propagates as a coastal Kelvin wave (Horner-Devine et al., 2015), leaving the coast

to the left in the southern hemisphere. This northward flow of the La Plata River plume is known as the BCC (De Souza and Robinson, 2004), stronger during a SW wind regime. Although the UCTD data suggests a strong stability on the continental shelf, values of $Ri_g$ are reduced when reaching the shelf break. The temperature data showed a vertical motion that appears to be related to the mechanism described by Wang (1984). In a rotating fluid over a slopping bottom, when two different water masses form a sharp density front, it stretches out from its initial position due to a reduction in the bottom layer velocity due

to friction, increasing the relative velocity in the top layer. Assuming that no mixing occurs, the density structure is a layer of lighter water overlaying a layer of heavier water. The resulting velocity distribution is a relatively strong recirculation, with sinking at the head of the surface front and rising at the head of the bottom front. In the observations, however, there is no indication of vertical intrusion, since the density contours remain partially stable, but vertical stretching is observed (c.f., Fig. 2). A baroclinic alongshore jet can be generated due to the geostrophic adjustment at the front. This is consistent with the

increased $\partial v/\partial z$ at the offshore limit of the plume and the fact that the La Plata River plume advection is nearly geostrophic (Pimenta et al., 2005). The occurrence of this mechanism at the bulge of the plume, the region where the river water outflow





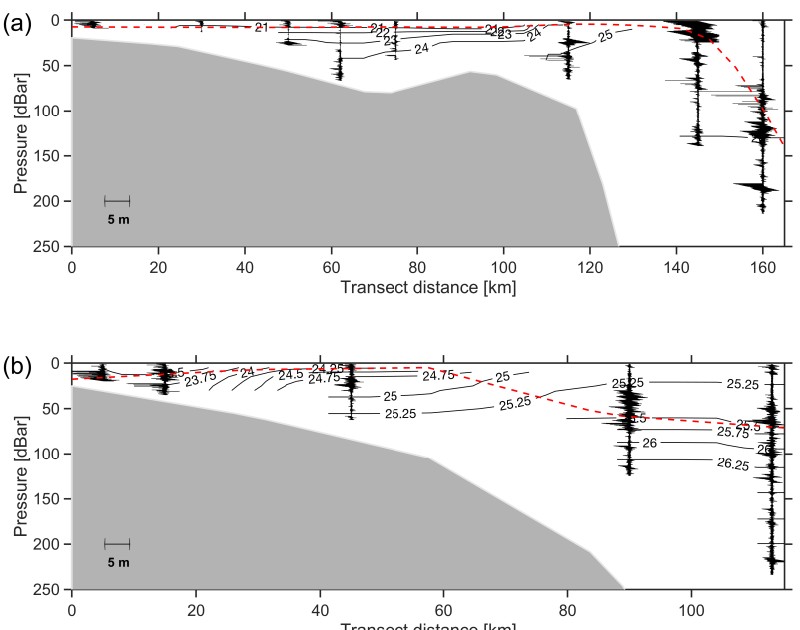

**Figure 10.** Density displacements calculated from the observed instantaneous microscale density profiles from the VMP-250 for the AL (a) and SM (b) transects. The black lines are the density contours in $kgm^{-3}$ and the red dashed line is the SML depth.

transitions into a geostrophic or wind-dominated far-field, suggests that the La Plata River plume owns a mid-field region, which is not usually expected for large-scale plumes.

Therefore, results show that plume stratification inhibits larger mixing at the Albardão region. Further north in the Sta. Marta region reduced stratification facilitates the generation of shear-stratified turbulence, increasing levels of diffusivity at the plume
5   boundary. Moreover, our observations suggest the presence of a mid-field region of mixing.

## 4   Conclusions

Our results show how the large-scale La Plata River plume affects vertical mixing in the shallow and dynamic SBS. Our observations are limited to two cruises of opportunity during Autumn conditions. However, two important conclusions may be drawn: (1) the transects in the SBS represent two distinct scenarios, as one was made close to the source region of the plume,
10   where stratification due to freshwaters is very strong, and the other made hundreds of kilometres away from it, where the freshwater outflow is more diluted with oceanic waters. In this sense, both show differences regarding shear-induced variability in the interface with oceanic waters, expressed by the density displacements and magnitudes of heat diffusivity. Near the source region, the plume is highly stable, while away from the plume, going northward, shear instabilities are significantly increased; (2) the instability observed in the hydrographic data close to the shelf break, associated with low values of gradient Richardson



number, suggests that the large-scale La Plata plume has a mid-field region of turbulence mixing, besides the far-field region located in the vicinity of Sta. Marta Cape. At the mid-field, data suggests that instabilities due to overturns in a sloping bottom is a source of turbulence, while in the far-field, KH instabilities are the causes of larger $K_T$.

Given the limitations imposed by the logistic of opportunity cruises and the limitations of our observations, this study is

an initial contribution to the knowledge of the La Plata River plume dynamics and provide a good base for future turbulence studies in this important region in the southwestern Atlantic Ocean. More detailed surveys in the area are necessary in order to fully characterize the turbulent mixing of the La Plata River plume. Future investigations may include:

- – In order to verify closely the possible implications of the BCC on the turbulent pattern in the coastal areas of the SBS, data must be obtained in areas between the AL and SM transects during SW winds regime. This would allow us to
determine how this current can alter the shape of the turbulence within the plume;

- – Obtain profiles of microstructure data near the entrance of the La Plata estuary, which could provide important information regarding the dynamics of the estuarine outflow at the source region;

- – Verify the implications of seasonal changes in the wind field and discharge rate of the La Plata estuary on the plume turbulence pattern;

– Although to assume $K_\rho$ similar to $K_T$ is a valid approximation in a turbulent scenario, it would be interesting to obtain estimates of the salt eddy diffusivity in the plume interface to complement the results obtained here;

- – Obtain measurements of $\epsilon$ from velocity shear variance, measured with shear probes (as those were not available during the survey herein) to estimate flux coefficient and mixing efficiency in the plume interface.

*Acknowledgements.* This study is part of the **ILHAS** project, which was supported by the Brazilian Government through the *Conselho*

*Nacional de Desenvolvimento Científico e Tecnológico* (CNPq) - Process 458583/2013-8 - Edital 62/2013 - Process 442926/2015-4 - Edital 15/2015. This study is also part of **REMARSUL** project, supported by *Coordenação de Aperfeiçoamento de Pessoal de Nível Superior* (CAPES) - Process 23038.004299/2014-53 - Edital *Ciências do Mar* II. PHRC acknowledges support from CNPq (Bolsa de Produtividade em Pesquisa - Process 306971/2016-0). RAA acknowledges support from Programa de Recursos Humanos (PRH-27) from Agência Nacional do Petróleo (ANP) through Ph. D. fellowship. We also thank Prof. Gilberto Henrique Griep for the participation in the cruises and the crew

and captain of R/V Atlântico Sul for valuable assistance.



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
