# Peer review of "BUOYANCY-DRIVEN EFFECTS ON TURBULENT DIFFUSIVITY INDUCED BY A RIVER PLUME IN THE SOUTHERN BRAZILIAN SHELF"

_Ocean Science, 2018_

## Referee Comment (RC1) · Anonymous Referee #1 · 24 Jul 2018

General comments: The manuscript by Avila and Calil presents an observational study of the Plata Plume Water on the Southern Brazilian Shelf. The manuscript does not pose any scientific gaps, research questions or research objectives that the work is addressing. It essentially presents some field observations of microstructure measurements in a case study. The overall organization of the manuscript needs to be restructured. It does not offer a discussion or summary section, nor does it present the organization of the paper in the introduction. There are grammatical errors throughout the paper that need to be addressed. I have serious concerns about the methodology behind some of the calculations used in this work, which I will detail below. The authors also seem to use estuarine references in support of river plume patterns, which

is not appropriate. These are considerable flaws and I do not think this manuscript is publishable in current form. I recommend rejection. However, I acknowledge the value of these data and believe that the authors could turn this into a valuable contribution. I encourage the authors to take this critical feedback and repackage these data into a manuscript that fills a targeted scientific gap, informed by a more thorough literature review.

Scientific issues: In measuring turbulence in river plumes, particularly in areas with Kelvin Helmholtz instabilities, the prevalence of intermittency needs to be addressed. Researchers typically average repeated casts to ensure the profile they collect isn't some rare intermittent event. There is no mention of repeated casts, or the total number of casts in this manuscript. I suggest reading MacDonald et al. 2013 for more info on intermittency. You cannot use inferred velocities from CTD measurements to calculate gradient Richardson numbers. Turbulence can locally impact the vertical distribution of velocity, which would not be captured by velocities inferred by using the thermal wind balance. Using this balance is fine for obtaining an order of magnitude idea of velocities, but you cannot use this to address mixing from shear instabilities in a river plume. This is probably why your gradient Richardson number are so large ($\sim$100). The important threshold to assess mixing is 0.25<Ri<1. Everything above Ri=3 has been shown to be converted to internal wave energy. The manuscript mentions that the instrument operated in down cast modes ranging from 0.2 to 1.4 m/s. The only portion of the water column that can be used is when the instrument is descending at a constant rate. This is because the instrument decent speed is used to convert to wavenumber space. Your descent speed should not be varying this much and would introduce errors in your turbulence calculations. Also, there is no mention of angle of attack. Typically, you need to remove bins where the instrument's angle from vertical is larger than +- 5 deg. You need to show example spectra and profiles with confidence intervals in order to provide evidence of the quality of the data. The usage of Peters (1997) and Stacey et al. (1999) to support claims regarding river plumes is perplexing. These are turbulence papers in estuaries, not river plumes and are not appropriate references. Matar et al. (2013) did not develop the theory behind Thorpe displacements. Reference the Thorpe text book. I do not understand where the double diffusion analysis comes into this work. How does this relate to river plume mixing? Your conclusions aren't actually conclusions, they are site specific patterns. First, you mention that the source region is stable and shear instabilities are increased farther away. Second, you point out mid field and far field regions of the plume and point out some mixing mechanisms. How do these observations fit into the broader understanding of river plumes? Are these patterns unexpected? Or would they be anticipated from previous studies?

A few minor comments: There are many grammatical errors but given the current state of this manuscript I am not going to go through them. There are way too many acronyms to keep track of in this work. Data is plural. You units are incorrect in the molecular thermal diffusivity Your references to patterns in figures needs to be more quantitative. Say notable values in text.

---

## Referee Comment (RC2) · Anonymous Referee #2 · 26 Jul 2018

This paper presents the results of hydrographic and turbulence measurements obtained along three transects off the southern coast of Brazil in the La Plata River plume region. Overall, the manuscript provides an interesting data set from a region that has not been well sampled in the past. Thus, the data alone are a significant contribution. The plume exits the La Plata river estuary and turns north due to Coriolis along the coast, in a pattern typical of large scale, far-field plumes. In this regard, the authors would be advised to compare the La Plata plume to similar plumes, such as the Chesapeake, or Delaware plumes, where significant recent work has been conducted. The authors conclude that the presence of the plume water, and the resulting stratification, reduces vertical mixing within the mixing within the plume, and the presence

of increasingly aged/mixed plume water at the transect furthest to the north, both expected conclusions. However, I encourage the authors to push further with the data to quantify more detailed aspects of the La Plata plume evolution. In summary, I believe the manuscript will ultimately be worthy of publication, but recommend major revisions to strengthen the scientific contribution of the work. A series of specific notes and comments follows:

Page 2, Line 2: Please explain the mechanisms by which increased frontal mixing results from plume deceleration in the mid-field region.

Page 3, Line 14: Ultimately, Rig is estimated strictly from density data by utilizing the thermal wind balance relationship. It would be instructive to include a revised equation for Rig based strictly on rho (i.e., combining the equation shown with the thermal wind balance equation). It should be noted, that this approach is not likely to provide enough resolution to use Rig as a diagnostic tool in evaluating the turbulent field. However, it may provide a useful context for the overall hydrography, but a clear representation of the fact that your Rig is approximated using thermal wind balance assumptions is necessary, which would be the result of representing the equation as suggested.

Page 12, Line 14 – Page 13, Line 2: I do not follow the argument that the presence of a baroclinic jet leads to the conclusion of a mid-field region. As described in Horner-Devine et al (2015) the mid-field region represents the transition from a non-geostrophic near-field to fully geostrophic far-field. Please clarify the intent in this section, and revise as necessary.

Section 3: As mentioned above, the manuscript could be improved by adding an additional sub section to Section 3, or a new section 4 for Analysis and Discussion. Most notably it appears that the two transects at opposing ends of the La Plata plume could be used (with some assumptions) to make some generalizations about La Plata plume evolution, including length/time scales for the erosion of stratification and the increase of mixing, allowing the ability for some comparisons with other similar plumes, such as

the Chesapeake and/or Delaware.

---

## Referee Comment (RC3) · Anonymous Referee #3 · 5 Aug 2018

The manuscript presents new observations in a poorly surveyed region, focusing on La Plata River Plume, Southwestern Atlantic Ocean. The data is certainly worthwhile of exploration due to its uniqueness and inherent value, but the manuscript at present form is not ready for publication. First, a goal was not clearly pursued: I felt that the paper was limited to simply presenting statistics of the data and did not provide a meaningful discussion, with insights on the broader significance of the observations for the dynamics of the region and how that fits into the existing literature, regional or from elsewhere. Second, the authors did not provide thorough information about the dataset and analysis: how exactly was the turbulence data set collected? how many profiles were taken? Certainly, more detail needs to be provided so the reader can

be assured that turbulence was resolved at the scales the authors claim they resolve. Third, there are major problems regarding the analysis and methods. For example, velocities estimated using thermal wind balance are not a proper way to estimate vertical shear for Richardson Number calculations, as by definition, they only account for the geostrophic portion of the flow, missing out the ageostrophic part, which could be dominant. If the authors want to estimate Richardson Numbers with field observations, they should obtain velocities using ADCPs. Finally, the text is poorly written in terms of structure and grammar, while it did not properly acknowledge the relatively large body of literature in the region (e.g. Matano, Combes, Palma, etc).
* * *